# CamoVid60K: A Large-Scale Video Dataset for Moving Camouflaged Animals Understanding

## Abstract

We have been witnessing remarkable success led by the power of neural networks driven by a significant scale of training data in handling various computer vision tasks. However, less attention has been paid to monitoring the camouflaged animals, the masters of hiding themselves in the background. Robust and precise segmentation of camouflaged animals is challenging even for domain experts due to their similarity to the environment. Although several efforts have been made in camouflaged animal image segmentation, to the best of our knowledge, limited work exists on camouflaged animal video understanding (CAVU). Biologists often prefer videos for monitoring and understanding animal behaviors, as videos provide redundant information and temporal consistency. However, the scarcity of labeled video data significantly hinders progress in this area. To address these challenges, we present **CamoVid60K**, a diverse, large-scale, and accurately annotated video dataset of camouflaged animals. This dataset comprises **218** videos with **62,774** finely annotated frames, covering **70** animal categories, which *surpasses* all previous datasets in terms of the number of videos/frames and species included. **CamoVid60K** also offers more diverse downstream tasks in computer vision, such as camouflaged animal classification, detection, and task-specific segmentation (semantic, referring, motion), *etc*. We have benchmarked several state-of-the-art algorithms on the proposed **CamoVid60K** dataset, and the experimental results provide valuable insights for future research directions. Our dataset serves as a novel and challenging benchmark to stimulate the development of more powerful camouflaged animal video segmentation algorithms, with substantial room for further improvement.

## 1 Introduction

The continuous evolution of neural networks (*e.g.*, Convolutional Neural Networks (CNNs) (He et al., 2016) and Vision Transformers (ViTs) (Dosovitskiy et al., 2020)) has provided powerful and efficient tools for visual understanding based on captured images and videos. Enhancements in both *data* and *algorithm* have led to significant progress and success in the field. Large-scale datasets (*e.g.*, COCO (Lin et al., 2014), ADE20K (Zhou et al., 2017) and Object365 (Shao et al., 2019)) with supervised annotations serve as essential stimuli for developing powerful visual perception algorithms (Xie et al., 2022) and benchmarking them to reveal future research directions. However, most existing datasets mainly contain everyday objects (*e.g.*, 80 categories in COCO). This work focuses on camouflaged animals, a less explored area of research. In addition, monitoring and understanding camouflaged animals is crucial for biodiversity conservation (Rands et al., 2010; Soofi et al., 2022), as it helps protect species that are otherwise difficult to detect and are at risk of unnoticed population declines. Furthermore, studying camouflaged animals provides insights into evolutionary biology and adaptation mechanisms, enriching our scientific understanding of natural selection.

However, unlike everyday objects, collecting images and videos of camouflaged animals is more challenging, and annotation procedures usually involve domain experts. *Segmentation*, which involves generating precise masks for objects of interest, is a fundamental task in computer vision. Camouflaged animal segmentation helps accurately identify and isolate these animals from their backgrounds in images, facilitating detailed study and analysis. The resulting masks aid in gathering precise data on their behavior, habitat, and population dynamics, enhancing our overall understanding of their ecology (Troscianko et al., 2017; Lv et al., 2021). Recently, several efforts Xie et al. (2022); Cheng

Table 1: Comparison with existing video animal datasets. Class.: Classification Label, B.Box: Bounding Box, Motion: Motion of Animal, Pseudo OF: Pseudo-label Optical Flow, Expres.: Referring Expression. ***Note that***, MVK Truong et al. (2023) dataset mostly consists of *normal* marine animals with only some camouflaged animals. The frequency of annotations refers to how often each frame is annotated. For instance, MoCA-Mask provides annotations for **every 5 frames**, resulting in only 4,691 annotated frames of 22,939 frames. In contrast, our CamoVid60K dataset offers a significantly larger volume of data with more frequent annotations and a wider variety of annotation types.

| Dataset | Venue | # videos / frames | # species | Frequency | Class. | B.Box | Mask | Motion | Pseudo OF | Expres. |
|---|---|---|---|---|---|---|---|---|---|---|
| CAD | ECCV'16 | 9 / 839 | 6 | every 5 frames | ✓ | | ✓ | | | |
| MoCA | ACCV'20 | 141 / 37,250 | 67 | **every frames** | ✓ | ✓ | | ✓ | | |
| MoCA-Mask | CVPR'22 | 87 / 22,939 | 44 | every 5 frames | ✓ | | ✓ | | | |
| MVK | MMM'23 | **1379 / ∼ 992,880** | - | every 30 frames | ✓ | | | | | ✓ |
| **CamoVid60K** | - | 218 / 62,774 | **70** | **every frames** | ✓ | ✓ | ✓ | ✓ | ✓ | ✓ |

et al. (2022b); Lamdouar et al. (2023); Vu et al. (2023) have been made to perform camouflaged animal segmentation. Specifically, camouflage is a powerful biological mechanism for avoiding detection and identification, making it more challenging to achieve precise segmentation.

Various datasets (*e.g.*, CAMO-COCO Le et al. (2019), COD10K Fan et al. (2022), CAM-LDR Lv et al. (2023), S-COD He et al. (2023b)) have been collected for image-level camouflaged animal segmentation. However, image-level camouflaged animal segmentation cannot fully satisfy biological monitoring and surveying purposes, where the activity and behavior (Yang et al., 2021) should be recorded. For video level, the MoCA dataset Lamdouar et al. (2020) is the most extensive compilation of videos featuring camouflaged objects, yet it only provides detection labels. We argue that bounding box annotations alone cannot adequately delineate camouflaged animals, especially those with irregular boundaries, poses, and patterns (*e.g.*, the transparent fins of fish). Furthermore, despite the shift from images to videos, the data annotations remain insufficient in both volume and accuracy for developing a reliable video understanding model capable of effectively handling complex camouflaged situations.

To fill this gap and advance camouflaged animal video understanding (CAVU) in real-world scenarios, we present **CamoVid60K**, a comprehensive video dataset dedicated to studying camouflaged animals. It comprises **218** videos with **62,774** finely annotated frames, covering **70** animal categories. Table 1 compares our proposed dataset with previous ones (CAD (Pia Bideau, 2016), MoCA (Lamdouar et al., 2020), MoCA-Mask (Cheng et al., 2022b), MVK (Truong et al., 2023)), showing that **CamoVid60K** *surpasses* all previous datasets in terms of the number of videos, frames, and species included. Unlike previous datasets that annotated every 5 frames, our dataset offers annotations for every single frame. Additionally, we provide **a wider variety of annotation types** (animal categories, bounding boxes, annotated masks, pseudo-label optical flow, referring expressions), making it a more effective benchmark for CAVU tasks. Our dataset supports **a broad range of downstream tasks**, as shown in Figure 1, including classification, detection, segmentation (semantic, referring, motion), and optical flow estimation, *etc*.

We propose baselines for each task and corresponding benchmarks to explore the capabilities of advanced algorithms in performing robust and precise video understanding. Our **CamoVid60K** serves as a novel and important testing set for both the computer vision and wildlife research communities.

Our main contributions are summarized as follows:

- We present a **large-scale** and **comprehensive** video dataset dedicated to the understanding of camouflaged animals, featuring **significantly more** data and annotation types than existing datasets.

- We propose a **simple pipeline** for camouflaged animal detection and segmentation that achieves performance comparable to state-of-the-art methods.

- We benchmark **various** camouflaged animal video understanding tasks, including image classification, object detection, and motion segmentation using several state-of-the-art models.

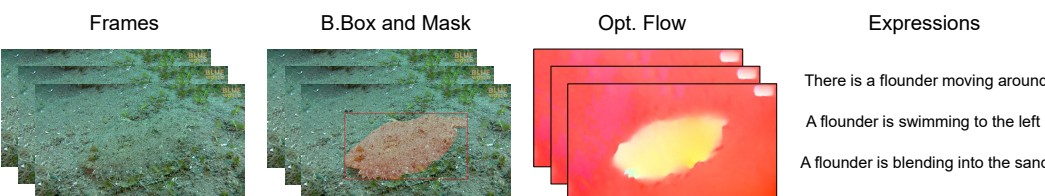

Figure 1: Example from our proposed **CamoVid60K** dataset with bounding box, mask, pseudo-label optical flow, and referring expressions.

## 2 RELATED WORKS

### 2.1 CAMOUFLAGED SCENE UNDERSTANDING

Camouflaged scene understanding (CSU) is a hot computer vision topic aiming to learn discriminative features that can be used to discern camouflaged target objects from their surroundings (Fan et al., 2023). CSU tasks can be divided into image-level and video-level categories. Image-level CSU tasks include five main types: camouflaged object counting (Sun et al., 2023), camouflaged object localization (Lv et al., 2021; 2023), camouflaged object segmentation (Fan et al., 2022; Ji et al., 2023; He et al., 2023a), camouflaged instance ranking (Lv et al., 2021; 2023), and camouflaged instance segmentation (Pei et al., 2022; Le et al., 2021). These tasks can be further categorized based on their semantic focus: object-level and instance-level. Object-level tasks focus on identifying objects, while instance-level tasks aim to differentiate various entities. Additionally, camouflaged object counting is considered a sparse prediction task due to its nature, while the other tasks are classified as dense prediction tasks. In addition, CSU video-level task includes video camouflaged object segmentation (Ji et al., 2014; Xie et al., 2019; Cheng et al., 2022b) and video camouflaged object detection (Lamdouar et al., 2020; 2021; Yang et al., 2021; Xie et al., 2022; Meunier et al., 2022; Kowal et al., 2022). Overall, the progress of video-level CSU has been somewhat slower than image-level CSU, primarily because the process of collecting and labeling video data is labor-intensive and time-consuming.

### 2.2 VIDEO CAMOUFLAGED OBJECT DETECTION AND SEGMENTATION

We review two kinds of perception tasks for camouflaged animal videos: detection (Lamdouar et al., 2020; 2021; Yang et al., 2021; Xie et al., 2022; Meunier et al., 2022; Kowal et al., 2022) and segmentation (Ji et al., 2014; Xie et al., 2019; Cheng et al., 2022b; Lamdouar et al., 2023). The former video camouflaged object detection (VCOD) yields bounding box sequences for the camouflaged animals, while the latter video camouflaged object segmentation (VCOS) generates dense pixel-level masks. MoCA Lamdouar et al. (2020) proposed the first large-scale moving camouflaged animals video dataset with bounding box annotations and additional optical flows to boost the detection of camouflaged animals. Further work Lamdouar et al. (2021) incorporated visual appearance from a static scene as additional clues to promote the ability of the model to detect camouflaged animals. However, the bounding box annotations could not accurately describe camouflaged objects' pose, appearance, and patterns. To address this issue, Xie Xie et al. (2019) proposed a novel pixel-trajectory RNN to cluster fore-ground pixels and generate dense segmentation masks for object discovery in videos. MoCA-Mask Cheng et al. (2022b) proposed the first large-scale dataset and benchmark with pixel-level handcrafted ground truth masks for camouflaged animal videos. However, MoCA-Mask provides bounding boxes and pixel-wise masks for **only every 5 frames**, totaling just 4,691 frames, which is insufficient for deep learning approaches. In contrast, our dataset offers annotations for **every frame**, resulting in 62,774 annotated frames (**13 times larger**). This substantial increase can significantly enhance the performance of various downstream tasks. Our dataset and benchmark pave the way for future exploration and a deeper understanding of camouflaged animal analysis.

## 3 CAMOVID60K DATASET

Collecting video datasets of camouflaged animals is quite challenging, even without focusing on long-form videos. Manually collecting, observing, and annotating videos with multiple annotation types is labor-intensive, time-consuming, and expensive. In addition to these costs, ensuring visual data diversity and high-quality annotations adds to the difficulty. In this section, we propose a

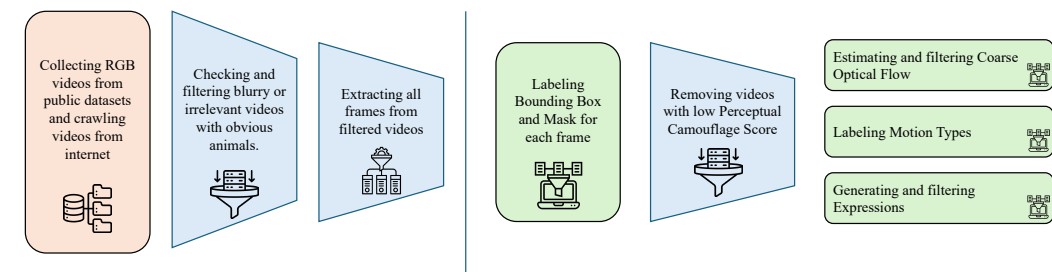

Stage I: Data curation and Filtering        Stage II: Annotations

Figure 2: **CamoVid60K** data pipeline. Stage I includes data curation, filtering irrelevant videos, and extracting all frames. Stage II includes data annotation, generation, and filtering.

staged data collection and processing pipeline, as shown in Figure 2. Associated datasheets (Gebru et al., 2021) and data cards (Pushkarna et al., 2022) for our **CamoVid60K** dataset are provided in Appendix B.

### 3.1 DATA CONSTRUCTION AND PROCESSING

**Data Sources and Pre-Processing.** We built our dataset by incorporating previous datasets (Table 1) and crawling additional videos from the internet to cover a variety of camouflaged animals. We initially collected 1,929 videos, then manually checked and filtered out any blurry or irrelevant videos, retaining those with clear depictions of animals. Next, we extracted every frame of each video (instead of every 5 frames as proposed in existing datasets, see Table 1) before annotating them. At the end, our dataset comprises **218** videos with **62,774** frames of **70** animal species. We provide more details in Appendix A.1.

**Bounding Box and Mask Annotation.** We utilized the annotation tool from (Zheng et al., 2023a), which is heavily based on the Segment Anything Model (SAM) (Kirillov et al., 2023) for mask initialization and bounding box creation, and XMem (Cheng & Schwing, 2022) for mask and bounding box propagation. We then manually checked and refined every frame to provide accurate bounding boxes and segmentation masks. In addition, we adopted the perceptual camouflage score ($\mathcal{S}_p$) from (Lamdouar et al., 2023) to quantify the effectiveness of animals' camouflage, *i.e.*, how successfully an animal blends into its background. Based on this score, we retained videos with a score higher than a threshold (*e.g.*, $\mathcal{S}_p > 0.3$).

*Note that,* due to the nature and characteristics of camouflaged animals and also the low resolution of videos, some frames or videos may contain errors or mislabeling at the boundaries between animals and the background. We will continue improving the quality of the mask annotations and also provide rotated bounding boxes (RBbox) in the next version. RBbox excels over traditional axis-aligned bounding boxes in three main areas: better localization (accurate fit for elongated and rotated objects), reduced overlap of different objects or instances, and improved isolation of objects (capturing the proper aspect ratio and containing fewer background pixels).

**Pseudo-label Optical Flow Annotation.** Previous optical flow datasets, such as Flying Chairs (Dosovitskiy et al., 2015), KITTI (Menze & Geiger, 2015), Sintel (Butler et al., 2012), and FlyingThings3D (Mayer et al., 2016), utilized either simulation software or real images with additional heavy sensor information (depth, LiDAR, *etc*.) and algorithms to create optical flow ground truth. This process is time-consuming and requires significant effort. Recently, with the development of deep learning techniques, many methods (Teed & Deng, 2020; Wang et al., 2023) can produce accurately estimated optical flow. Therefore, we utilized these methods for our pseudo-label optical flow annotation, using the algorithm shown in Algorithm 1. We used the pre-trained model of RAFT on FlyingThings3D (Mayer et al., 2016) and the pre-trained DINO model of ViT-B architecture.

*Note that,* even though our processing pipeline for optical flow annotation produces accurate and dense optical flow, it is still **estimated** optical flow. Therefore, it is reasonable and suitable to use as *additional input* to boost performance for other tasks such as motion segmentation. It is **not recommended** to use it as ground truth for evaluation.

---

**Algorithm 1** Optical Flow Computation and Filtering

---

**Input:** Sequence of frames
**Output:** Sequence of computed optical flows
 1: **for** each pair of frames $(i, j)$ **do**
 2:     Computing all pairwise optical flows using RAFT (Teed & Deng, 2020)
 3:     Computing DINO features (Caron et al., 2021; Oquab et al., 2024) for each frame
 4:     Filtering flows using cycle consistency and appearance consistency check
 5:     Applying chain cycle consistent correspondences to create denser correspondences
 6: **end for**

---

**Motion Annotation.** Following Lamdouar et al. (2020), we manually labeled our dataset according to the types of motion, as shown below. Based on these motion types, we can further annotate the camouflage methods of animals (concealing coloration, disruptive coloration, disguise, mimicry, transparency, and counter-shading), which we plan to provide in the next version.

- *Locomotion*: when the animal makes movements that significantly change its location.

- *Deformation*: when the animal engages in more subtle movements that only change its pose while remaining in the same location.

- *Still*: when the animal remains stationary.

**Referring Expression Annotation.** We first utilized GPT-4V (OpenAI, 2023) to create concise descriptions within 30 words that accurately represent the target object for every frame. However, we found that the captions for aquatic animals were less accurate; therefore, we utilized MarineGPT (Zheng et al., 2023b), the first vision-language model specially designed for the marine domain, to generate captions for aquatic animals. After the initial annotation, we verified and refined all captions and selected the best three for each video sequence. Objects that could not be localized using language or referring expressions were removed. We provide more details on the definition and usage in Appendix A.3.

### 3.2 DATASET SPECIFICATIONS AND STATISTICS

**Data Organization.** As shown in Figure 3, we split our dataset into two subsets based on the degree of displacement between frames: small displacement (every single frame) and large displacement (every 5 frames). This division is beneficial for evaluating motion segmentation methods, as it provides a robust framework for analyzing algorithms' performance under varying motion and displacement conditions. Each subset includes training and testing sets with images, pre-computed optical flows, and annotations. We name each image using the following format:

```
CamoVid60k
|
├── Small_Displacement
|   ├── Train_Set (168 videos)
|   |   ├── Aquatic-Crab-3
|   |   |   ├── Images (Aquatic-Crab-3-Loco-0001.jpg)
|   |   |   ├── Flows (Aquatic-Crab-3-Loco-0001.flo)
|   |   |   └── Annotations
|   |   |       ├── annotations.json (bounding boxes and masks)
|   |   |       └── exp_annotations.json (expressions)
|   |   ├── Terrestrial-Wolf-1
|   |   └── Flying-Owl-1
|   └── Test_Set (50 videos)
|
└── Large_Displacement
    ├── Train_Set
    └── Test_Set
```

Figure 3: Data organization of our dataset.

`"SuperClass-SubClass-SubNumber-MotionType-FrameNumber"`. This systematic naming convention ensures clarity and ease of reference within the dataset.

**Dataset Features and Statistics.** We now discuss the proposed dataset and provide some statistics.

- *Category diversity:* The distributions of camouflaged animals, categorized hierarchically based on biology within three supergroups (flying, terrestrial, and aquatic), are visually represented through

taxonomic structures in Figure 4 (Left) and word clouds in Figure 5. Subsequently, we describe the 70 prevalent subclass groups derived from our collected data in Figure 6.

- *Spatial distribution of animals' positions:* Figure 4 (Right-Top) and Figure 5 (Bottom) showcase examples with different animal positions and present the total sum of normalized bounding boxes across the entire dataset.

- *Resolution distribution:* Using high-resolution data is beneficial as it offers more detailed object boundary information for model training, thereby improving performance during testing (Zeng et al., 2019). In Figure 4 (Right-Bottom), the resolution distribution of **CamoVid60K** is displayed, highlighting the inclusion of numerous HD (720p) and Full HD (1080p) resolution videos.

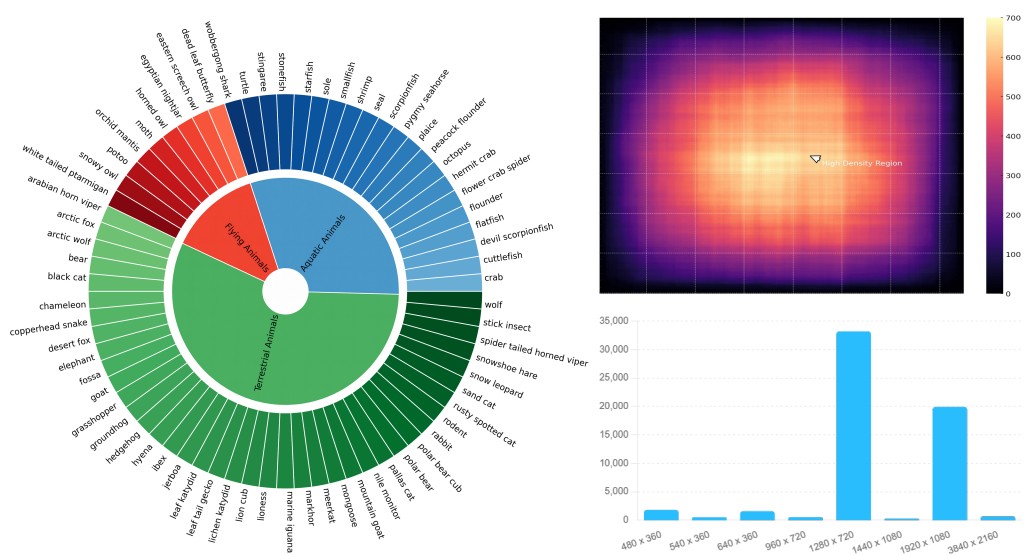

Figure 4: **Left:** Taxonomic structure of our dataset by their biology-inspired hierarchical categorization. It encompasses various animals, spanning 70 categories across flying, terrestrial, and aquatic groups. **Right-Top:** Spatial distribution of animals' position based on bounding box. It reveals that annotations are more densely concentrated in the central region, while there is a comparatively lower density of annotations towards the edges. **Right-Bottom:** The distribution of our CamoVid60K dataset w.r.t resolution ranging from 480×360 to 3840×2160.

**Evaluation Protocol.** Our dataset supports a broad range of downstream tasks. Therefore, we will evaluate each task using different metrics.

- *Motion Segmentation:* we adopt the same metrics as in (Cheng et al., 2022b) to assess the pixel-wise masks: Mean Absolute Error ($M$), Enhanced-alignment measure ($E_\phi$) (Fan et al., 2018), Structure-measure ($S_\alpha$) (Fan et al., 2017), Weighted F-measure ($F_\beta^w$) (Margolin et al., 2014), mean Intersection Over Union (mIoU), mean Dice (mDic).

- *Object Detection:* we use the mean Average Precision (mAP).

- *Image Classification:* we use the mean Accuracy (mAcc).

- *Referring Segmentation:* we utilize the mIoU, region similarity $\mathcal{J}$ and contour accuracy $\mathcal{F}$, and their average $\mathcal{J}\&\mathcal{F}$ for video object segmentation.

## 4 A SIMPLE PIPELINE TO DISCERN CAMOUFLAGED ANIMALS

After constructing the dataset, we propose a simple pipeline based on the Mask2Former architecture (Cheng et al., 2022a; Lamdouar et al., 2023) for both object detection and motion segmentation tasks. As shown in Figure 7, our pipeline processes sequences of images or videos by employing any off-the-shelf flow estimation method. In our case, we directly use the refined optical flow provided in our dataset instead of utilizing the RAFT method (Teed & Deng, 2020) to estimate raw optical flow, as done in (Lamdouar et al., 2023). The images and associated estimated flows are passed into two

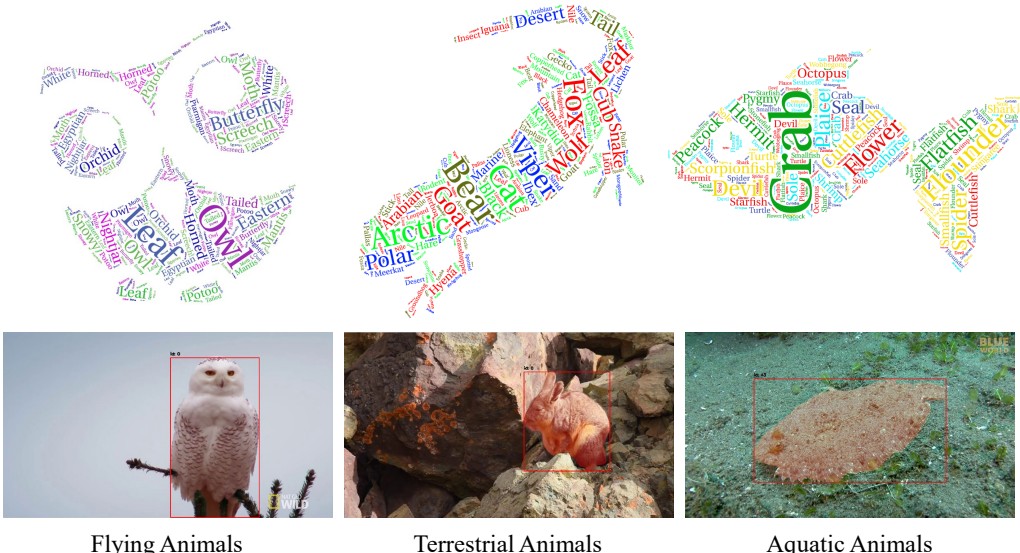

Flying Animals      Terrestrial Animals      Aquatic Animals

Figure 5: Word cloud of category distribution of camouflaged animals with corresponding examples showing bounding box, segmentation mask (bottom).

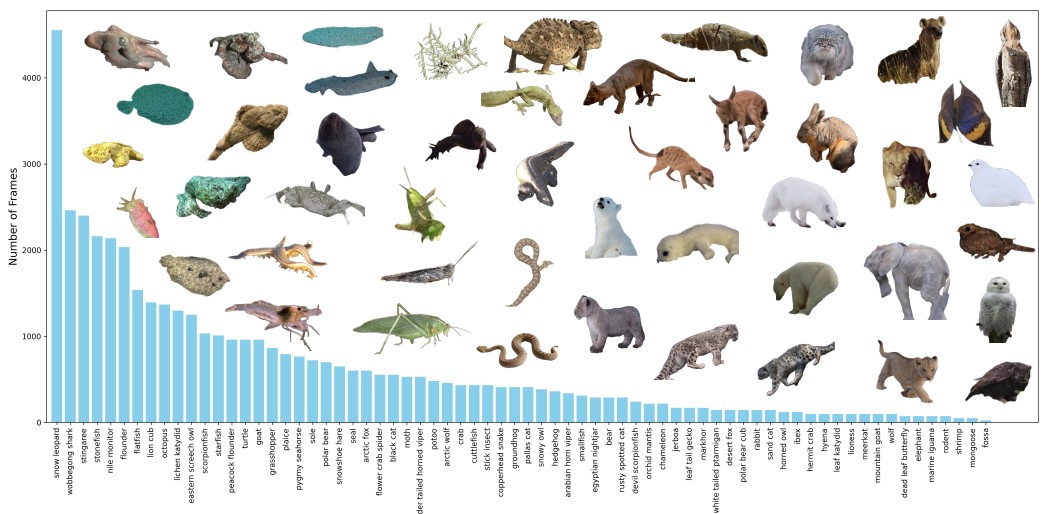

Figure 6: Category distribution (ranging from 100 to 4,500 frames) and some visual examples (extracted animal masks) of our dataset. The variety ensures a wide range of camouflaged animals, allowing for comprehensive evaluation across various scenarios.

separate encoders for feature extraction. Subsequently, the image and flow features at each timestamp are aggregated before being fed into the decoder to predict the segmentation mask.

**Visual Encoder.** We adopt the SINet-v2 (Fan et al., 2022) architecture, which takes an RGB sequence as input $I^v = \{I_1^v, I_2^v, \ldots, I_n^v\} \in \mathbb{R}^{n \times d_v \times h \times w}$, where $n$ is the number of frames, $d_v$ is the dimension of each frame, and $h$ and $w$ are the height and width, respectively. The visual encoder outputs visual features $\{f_1^v, f_2^v, \ldots, f_n^v\} = \Phi_{\text{visual}}(I^v)$.

**Motion Encoder.** Inspired by the motion segmentation architecture (Lamdouar et al., 2021), we use a lightweight ConvNet that takes as input a sequence of optical flows $I^f = \{I_1^f, I_2^f, \ldots, I_n^f\} \in \mathbb{R}^{n \times d_f \times h \times w}$, where $d_f$ is the dimension of the flow field, and outputs motion features $\{f_1^m, f_2^m, \ldots, f_n^m\} = \Phi_{\text{motion}}(I^f)$. We then concatenate the motion features with learned spatial and temporal positional encodings to produce a set of enriched motion features.

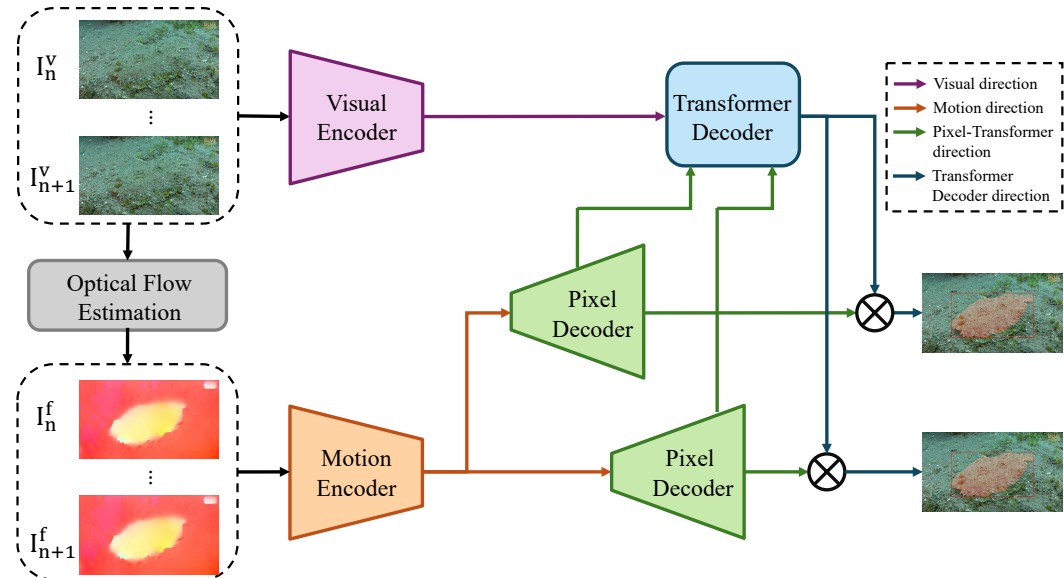

Figure 7: Our simple pipeline takes a sequence of images (or a video) and the associated pre-computed optical flow (provided in our dataset) as input. They are fed into separated encoders for feature extraction. Then, the motion features with spatial and temporal positional encoding are passed to Pixel Decoders to produce a set of enriched motion features. Next, the Transformer Decoder takes the visual features and enriched motion features to produce mask embedding for the moving object and bounding box.

**Decoder.** We adopt the Mask2Former (Cheng et al., 2022a) architecture, which includes Transformer and pixel decoders. The Transformer decoder combines a trainable query for mask embedding with the outputs of the motion encoder and visual features. Similar to Mask2Former, this query attends to multi-scale motion features and visual features, resulting in mask embedding for the moving object. Additionally, like the pixel decoder in Mask2Former, a ConvNet decoder with low computational complexity utilizes skip connections to generate high-resolution segmentation masks and bounding boxes from the motion features and mask embedding.

**Training and Loss.** To optimize our pipeline, we utilize the L1 loss for bounding box regression, cross-entropy loss for the confidence score, and binary cross-entropy (BCE) loss for motion segmentation. The total loss for training our pipeline is defined as follows:

$$\mathcal{L} = \mathcal{L}_{\text{BCE}} + \mathcal{L}_{\text{L1}} + \mathcal{L}_{\text{CE}}, \tag{1}$$

where $\mathcal{L}_{\text{BCE}}$ is the binary cross-entropy loss for motion segmentation, $\mathcal{L}_{\text{L1}}$ is the L1 loss for bounding box regression, and $\mathcal{L}_{\text{CE}}$ is the cross-entropy loss for the confidence score.

## 5 EXPERIMENTS

This section introduces the baselines and training details for each task. We thoroughly analyze each task in our experiments and discuss the effectiveness of each method, including ours.

### 5.1 BASELINES

**For the motion segmentation task**, we selected recent state-of-the-art (SOTA) methods for comparison, including two frame-based methods (PraNet (Fan et al., 2020) and SINet-v2 (Fan et al., 2022)) and two video-based methods (MG (Yang et al., 2021) and SLT-Net (Cheng et al., 2022b)). For a fair comparison, we utilized the implementations provided by the authors and trained all methods using the same training set.

**For the object detection task**, we compared our approach with three well-known detection methods: Faster R-CNN (Ren et al., 2015), DETR (Carion et al., 2020), and DINO (Zhang et al., 2023). We

Table 2: Quantitative results of motion segmentation on CamoVid60K dataset. Our model achieves performance comparable to that of other competitors on certain metrics.

Table 3: Quantitative results of object detection on our CamoVid60K dataset.

| Methods | | $S_\alpha \uparrow$ | $F_\beta^w \uparrow$ | $E_\phi \uparrow$ | $M \downarrow$ | mDic $\uparrow$ | mIoU $\uparrow$ |
|---------|---|---------|---------|---------|---------|---------|---------|
| Image-based | PraNet | 0.526 | 0.161 | 0.547 | 0.045 | 0.198 | 0.144 |
| | SINet-v2 | 0.529 | 0.166 | 0.553 | 0.042 | 0.206 | 0.149 |
| Video-based | MG | 0.522 | 0.153 | 0.541 | 0.043 | 0.197 | 0.141 |
| | SLT-Net | **0.576** | **0.253** | **0.591** | **0.039** | 0.268 | 0.249 |
| | Ours | 0.566 | 0.249 | 0.589 | 0.041 | **0.270** | **0.252** |

| Methods | $AP \uparrow$ |
|---------|---------|
| F-RCNN | 28.71 |
| DETR | 37.56 |
| DINO | **39.84** |
| Ours | 38.39 |

followed the $1\times$ (12-epoch) training setting and used the same ResNet50 (He et al., 2016) backbone for all methods.

**For the zero-shot image classification task**, we tested three recent methods: CLIP (Radford et al., 2021), UniCL (Yang et al., 2022), and K-LITE (Shen et al., 2022). We used the Swin-T model for both UniCL and K-LITE (pre-trained on the ImageNet-21K dataset (Deng et al., 2009)) and the ViT-B/32 pre-trained model from OpenAI's CLIP.

All methods were trained and tested on the same NVIDIA RTX 3090 GPU, except for the pre-trained models used in the zero-shot image classification task, where we utilized the pre-trained models provided by the authors.

## 5.2 BENCHMARKS AND DISCUSSIONS

**Comparison with Image-Based and Video-Based Motion Segmentation Methods.** Table 2 compares the performance of our method with other approaches. Compared to image-based methods, our approach demonstrates significantly superior performance due to the incorporation of temporal information. When evaluated against video-based methods, our approach also surpasses MG (Yang et al., 2021), which relies solely on estimated optical flows as input. However, compared to the recent state-of-the-art method SLT-Net (Cheng et al., 2022b), our method performs worse on certain metrics. This is because SLT-Net excels at modeling both short-term dynamics and long-term temporal consistency from videos, allowing for joint optimization of motion and camouflaged object segmentation through a single optimization target.

**Comparison with Object Detection Methods.** As shown in Table 3, our proposed model demonstrates performance comparable to other specialized methods, owing to its dual capabilities in object detection and motion segmentation. Specifically, our method significantly outperforms conventional CNN-based methods. This advantage stems from dual optimizations in the detection and segmentation streams, along with the integration of additional optical flow information. However, when compared to DETR-like methods (Carion et al., 2020; Zhang et al., 2023), our approach shows mixed results. It surpasses the standard DETR model (Carion et al., 2020), yet falls short of DINO (Zhang et al., 2023), an advanced variant of DETR. DINO enhances performance through several innovative techniques: it employs contrastive denoising training to refine one-to-one matching, a mixed query selection method to better initialize the queries, and a 'look forward twice' method that utilizes gradients from subsequent layers to adjust parameters more accurately.

**Additional Analysis and Discussions.** As shown in Table 4, optical flow plays a crucial role in the motion segmentation of camouflaged animals. By analyzing the motion vectors between frames, optical flow can detect subtle movements, distinguishing moving animals from static backgrounds. This capability is particularly useful in identifying the slight movements of camouflaged animals.

State-of-the-art methods, including foundation models trained on large datasets such as CLIP (Radford et al., 2021), UniCL (Yang et al., 2022), and K-LITE (Shen et al., 2022), struggle with zero-shot image classification of camouflaged animals, as shown in Table 5. This is due to the subtle and complex patterns of camouflaged animals, the lack of specific training data, and the difficulty in generalizing across different backgrounds and lighting conditions. Improving these methods involves

Table 4: Ablation study on the impact of flow information on our method.

|  | no OF | raw OF | refined OF |
|---|---|---|---|
| mIoU | 28.34 | 32.16 | **32.81** |

Table 5: Zero-shot Image Classification performance on our CamoVid60K dataset.

|  | CLIP | UniCL | K-LITE |
|---|---|---|---|
| mAcc | 30.06 | 30.89 | **31.44** |

curating specialized training data (or fine-tuning on our dataset), using enhanced techniques like data augmentation, few-shot learning, and developing context-aware models.

## 6 CONCLUSION

In this paper, we introduced **CamoVid60K**, a large-scale video dataset for camouflaged animal understanding, aiming to foster further research on camouflaged animals. This dataset provides a significant benchmark for camouflaged animal video understanding tasks, enabling the evaluation of various algorithms and methods. We also plan to scale up our dataset and utilize it to build foundational models for studying camouflaged animals.

**Limitations and Future Work.** As mentioned in Section 3, the annotation quality in some cases is suboptimal. We plan to enhance these annotations and introduce more types of annotations in the future. Additionally, our current pipeline requires images and pre-computed optical flow as inputs, which restricts the generation of new data due to the necessity of pre-computed optical flow. To address this limitation, we will propose a learnable module to estimate the implicit optical flow field.

**New Benchmark.** **CamoVid60K** is a diverse and comprehensive benchmark curated from publicly accessible datasets and the internet to enhance the assessment and exploration of camouflaged animal understanding. It includes various camouflaged animals across different environments, providing a robust framework for testing and developing new models.

**Impact on Animal Studies.** By providing detailed and varied data on camouflaged animals, the **CamoVid60K** dataset significantly contributes to studying animal behavior, ecology, and evolution. Researchers can utilize this dataset to explore how different species employ camouflage in their natural habitats, leading to deeper insights into predator-prey interactions and survival strategies. Furthermore, this dataset can aid conservation efforts by improving the detection and monitoring of endangered species in their natural environments (Troscianko et al., 2017; Norouzzadeh et al., 2018; Beery et al., 2018; Simões et al., 2023).

**Broader Impact.** The study of camouflaged objects has several important applications, such as identifying and safeguarding rare animal species, preventing wildlife trafficking, detecting medical conditions like polyps or lung infections, and aiding in search-and-rescue operations. Our dataset deliberately excludes any military or sensitive scenes, ensuring its focus remains on benign and beneficial applications. Besides the significant applications mentioned, our work advances the understanding of video content in the presence of distorted motion information, contributing to the broader field of video analysis and computer vision.

**Licenses.** We built our dataset from previous datasets and crawled online videos. Therefore, we will follow their Terms of Use or Licenses (MoCA, MVK) for our dataset, which is under the CC-BY-4.0 license. The copyright remains with the original owners of the videos. In addition, the dataset shall be used only for non-commercial research and educational purposes.

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

# A CAMOVID60K DESCRIPTION

## A.1 DATA CURATION

We built our dataset from published datasets (Camouflaged Animals Dataset (CAD) (Pia Bideau, 2016), Moving Camouflaged Animals (MoCA) (Lamdouar et al., 2020), MoCA-Mask (Cheng et al., 2022b), Marine Video Kit (MVK) (Truong et al., 2023)) and crawled video from internet.

The CAD dataset includes nine short video sequences obtained from YouTube videos. Hand-labeled ground truth masks are provided for every 5 frames.

The MoCA dataset comprises approximately 37,000 frames extracted from 141 YouTube video sequences. Most videos are presented at an image resolution of $1280 \times 720$ and $3840 \times 2160$ pixels, and the videos have a frame rate of 24 FPS. This dataset includes 67 distinct species of animals in locomotion within their native habitats, although it contains a few instances of animals with less camouflaged characteristics.

The MoCA-Mask dataset is built upon the MoCA dataset with some modifications. This new subset consists of 87 video sequences with 22,939 frames. It offers precise human-labeled segmentation masks for every 5 frames. Consequently, the ground truth (GT) is available in two formats: 4,691 bounding box annotations and 4,691 pixel-level masks.

The MVK dataset comprises 1,379 underwater videos recorded at 36 unique geographical sites during various seasons. These videos exhibit a broad duration spectrum, ranging from as short as 2 seconds to almost 5 minutes, with a total duration slightly above 12 hours. On average, the videos are roughly 29.9 seconds long, with a median length of around 25.4 seconds. Notably, the dataset presents videos recorded under different conditions, such as variable light levels, points of view, water clarity, and environmental conditions. They also offer approximately 40,000 frames (extracted at one FPS or every 30 frames) with associated referring expression annotations.

To crawl videos from the internet, we curated a list of animal names that potentially have camouflage abilities. We then created a template for searching and downloading videos: *"video of camouflaged/concealed + animal's name"*. Combining these with the videos from the above datasets, we collected 1,929 videos.

## A.2 DATA FILTERING

Initially, we manually reviewed and removed blurry or irrelevant videos (*e.g.* those with clearly visible animals), resulting in 218 videos for annotation. To further filter images and annotations with less camouflaged characteristics, we adopted the perceptual camouflage score ($\mathcal{S}_p$) from (Lamdouar et al., 2023) to quantify the effectiveness of animals' camouflage, *i.e.*, how successfully an animal blends into its background. Based on the perceptual camouflage score, we retained videos with a score

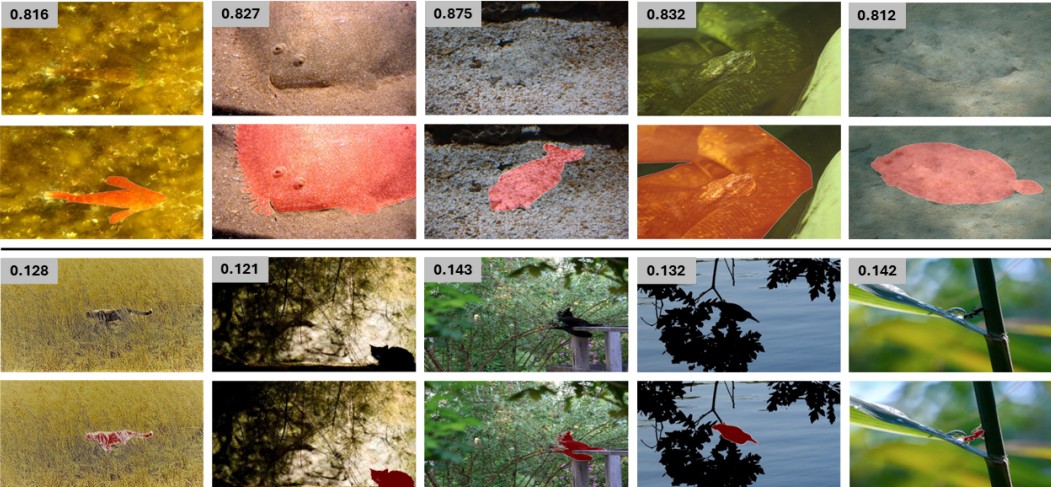

Figure 8: The example of low ranking and high ranking camouflage of single frame.

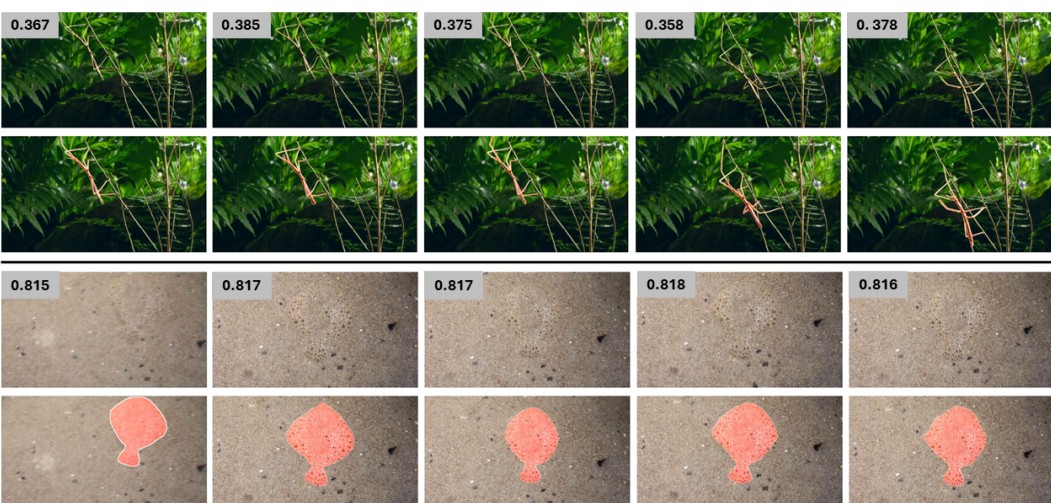

Figure 9: The example of low ranking and high ranking camouflage of consecutive frames.

higher than the threshold ($\mathcal{S}_p > 0.5$). Below, we explain how to compute the perceptual camouflage score $\mathcal{S}_p$:

$$\mathcal{S}_p = (1 - \alpha)\mathcal{S}_\mathcal{R} + \alpha\mathcal{S}_\mathcal{B} \tag{2}$$

where $\mathcal{S}_\mathcal{R}$ is the reconstruction fidelity score, $\mathcal{S}_\mathcal{B}$ is the boundary score, and $\alpha$ is the weighting parameter.

In detail, given an image $\mathcal{I}$ and a segmentation mask $m_S$, the reconstruction fidelity score $\mathcal{S}_\mathcal{R}$ is computed by assessing the difference between the foreground region and its reconstruction. Specifically, it counts the number of foreground pixels ($\mathcal{I}_{\text{fg}} = \mathcal{I} \odot \text{erode}(m_S)$) that have been successfully reconstructed from the background ($\mathcal{I}_{\text{bg}} = \mathcal{I} \odot (1 - \text{dilate}(m_S))$):

$$\mathcal{S}_{\mathcal{R}}(\mathcal{I}, m_S) = \frac{1}{N_{\text{fg}}} \sum_{(i,j) \in \mathcal{I}_{\text{fg}}} \mathcal{R}(i,j), \tag{3}$$

$$\mathcal{R}(i,j) = \begin{cases} 1, & \text{if } \left\| \mathcal{I}_{\text{fg}}(i,j) - \Psi_{\mathcal{I}_{\text{bg}}}(\mathcal{I}_{\text{fg}}(i,j)) \right\|_2 < \lambda \left\| \mathcal{I}_{\text{fg}}(i,j) \right\|_2, \\ 0, & \text{otherwise,} \end{cases} \tag{4}$$

where $\Psi_{\mathcal{I}_{\text{bg}}}(\cdot)$ denotes the reconstruction operation, $N_{\text{fg}} = |\text{erode}(m_S)|$ is the total number of pixels in the foreground region, and $\lambda$ is a threshold.

Then, the boundary visibility score $\mathcal{S}_{\mathcal{B}}$ aims to measure the animal's boundary properties (or contour visibility) by penalizing the boundary pixels that are predicted as contours in both the image's contour ($\mathcal{C}$) and the ground truth animal's contour ($\mathcal{C}_{\text{gt}}$) using the F1 metric:

$$\mathcal{S}_{\mathcal{B}}(\mathcal{I}, m_S) = 1 - \text{F1}(m_b \odot \mathcal{C}_{\text{gt}}, \ m_b \odot \mathcal{C}), \tag{5}$$

where $m_b = \text{dilate}(m_S) - \text{erode}(m_S)$.

We used the same parameter values as in (Lamdouar et al., 2023), specifically $\alpha = 0.35$ and $\lambda = 0.2$. In addition, we illustrate the difference between low-ranking and high-ranking camouflage in Figure 8 and Figure 9.

### A.3 REFERRING EXPRESSION ANNOTATION

Referring expression annotations are used for the Referring Video Object Segmentation (RVOS) task. RVOS differs from traditional Video Object Segmentation (VOS), where a mask is provided for the first frame, and the model predicts the segmentation for the remaining video frames. In RVOS, the initial frame mask is replaced with a referring expression (*i.e.* a sentence) that accurately describes the target object throughout the entire video, *e.g. "the yellow fish swimming toward the camera."* This approach also differs from Referring Image Segmentation (RIS), which uses different referring expressions for each image. Referring expression annotations can be utilized for various video understanding tasks, such as RVOS (Seo et al., 2020; Yang et al., 2024), video retrieval systems with semantic understanding (Ha et al., 2023), video grounding (Mu et al., 2024), *etc*.

### A.4 VISUALIZATION

We show some samples in alphabetical order of our CamoVid60K dataset in the attachment (index.html).

## B CAMOVID60K DATASHEET

| **Motivation** |
| :---: |

**For what purpose was the dataset created?** Was there a specific task in mind? Was there a specific gap that needed to be filled? Please provide a description.

There are some studies about camouflaged animal segmentation, and most of them are image-based methods. While some prior works have proposed video datasets for camouflaged animal understanding, they only provided a small amount of data with limited annotation types. To address those challenges and promote more studies on biological monitoring and understanding of animals' behavior, we introduce our CamoVid60K dataset and related benchmarks for a broad range of video understanding tasks. Please see Section 3 and Section 5 in the main paper for more details.

**Who created this dataset (*e.g.* which team, research group) and on behalf of which entity (*e.g.* company, institution, organization)?**

864
865
866
867

The authors created the dataset from the XXX and YYY Institutions. The authors created it for the public at large without reference to any particular organization or institution.

868
869

| **Composition** |
| --- |

870
871
872
873

**What do the instances that comprise the dataset represent (*e.g.* documents, photos, people, countries)?**  Are there multiple types of instances (*e.g.* movies, users, and ratings; people and interactions between them; nodes and edges)? Please provide a description.

874
875
876

Each instance in the dataset represents a sequence of extracted frames from a video with different annotations (category, bounding box, mask, motion type, pseudo-label optical flow, and three referring expressions.

877
878

**How many instances are there in total (of each type, if appropriate)?**

879
880
881

CamoVid60K has a total of 218 instances, each containing frames, associated bounding box, mask, motion type, pseudo-label optical flow, one category, and three referring expressions. You can see further statistics on the whole data in Section 3 of the main paper.

882
883
884
885
886
887

**Does the dataset contain all possible instances, or is it a sample (not necessarily random) of instances from a larger set?**   If the dataset is a sample, then what is the larger set? Is the sample representative of the larger set (*e.g.* geographic coverage)? If so, please describe how this representativeness was validated/verified. If it is not representative of the larger set, please describe why not (*e.g.* to cover a more diverse range of instances because instances were withheld or unavailable).

888
889
890
891
892

The dataset contains all instances from previous datasets with additional new data that are crawled from the internet to provide a larger volume of data with more frequent annotations and types and cover a wider variety of species ranging from flying to terrestrial and aquatic animals. The detailed statistics are shown in Table 1 and Section 3 of the main paper.

893
894

**What data does each instance consist of?  "Raw" data (*e.g.* unprocessed text or images) or features?** In either case, please provide a description.

895
896
897

Each instance in our dataset comprises raw mp4 video data, captured at 24-30 frames per second and with resolution from $480{\times}360$ to $3840{\times}2160$.

898
899

**Is there a label or target associated with each instance?**  If so, please provide a description.

900
901
902

Each instance is associated with a bounding box, mask, motion type, pseudo-label optical flow, one category, and three referring expressions.

903
904
905

**Is any information missing from individual instances?** If so, please provide a description, explaining why this information is missing (*e.g.*, because it was unavailable). This does not include intentionally removed information but might include, *e.g.* redacted text.

906
907

All instances are complete.

908
909
910
911

**Are relationships between individual instances made explicit (*e.g.* users' movie ratings, social network links)?** If so, please describe how these relationships are made explicit.

912
913

Some instances may have the same category name and similar referring expressions because they belong to the same category. However, each instance will have its unique ID.

914
915
916

**Are there recommended data splits (*e.g.* training, development/validation, testing)?** If so, please provide a description of these splits, explaining the rationale behind them.

917

CamoVid60K is explicitly designed for learning both small and large motion displacement of camouflaged animals.  Therefore, it is split into two subsets: small displacement (every single

frame) and large displacement (every 5 frames). This division is beneficial for evaluating motion segmentation methods, as it provides a robust framework for analyzing algorithms' performance under varying motion and displacement conditions. Each subset will include training (168 instances) and testing sets (50 instances), as mentioned in Section 3.2 of the main paper.

**Are there any errors, sources of noise, or redundancies in the dataset?** If so, please provide a description.

The dataset was carefully manually curated to mitigate any errors within the questions and answers. However, due to the nature and characteristics of camouflaged animals and their resolution, some frames will contain errors/mislabelled at the boundary between the animals and the background. We will keep improving the quality of the mask annotations in the next version.

**Is the dataset self-contained, or does it link to or otherwise rely on external resources (*e.g.* websites, tweets, other datasets)?** If it links to or relies on external resources, a) are there guarantees that they will exist, and remain constant, over time; b) are there official archival versions of the complete dataset (*i.e.* including the external resources as they existed at the time the dataset was created); c) are there any restrictions (*e.g.* licenses, fees) associated with any of the external resources that might apply to a future user? Please provide descriptions of all external resources and any restrictions associated with them, as well as links or other access points, as appropriate.

Entirety of the dataset will be made publicly available at our CamodVid60K website (we will update our website later). CamoVid60K will be publicly released under the CC-BY-4.0 license, which allows public use of the video and annotation data for both research and commercial purposes.

**Does the dataset contain data that might be considered confidential (*e.g.* data that is protected by legal privilege or by doctor-patient confidentiality, data that includes the content of individuals non-public communications)?** If so, please provide a description.

No

**Does the dataset contain data that, if viewed directly, might be offensive, insulting, threatening, or might otherwise cause anxiety?** If so, please describe why.

No

**Does the dataset relate to people?** If not, you may skip the remaining questions in this section.

No, CamoVid60K only contains animals.

**Does the dataset identify any subpopulations (*e.g.* by age, gender)?** If so, please describe how these subpopulations are identified and provide a description of their respective distributions within the dataset.

No

**Is it possible to identify individuals (*i.e.* one or more natural persons), either directly or indirectly (*i.e.* in combination with other data) from the dataset?** If so, please describe how.

No

**Does the dataset contain data that might be considered sensitive in any way (*e.g.* data that reveals racial or ethnic origins, sexual orientations, religious beliefs, political opinions or union memberships, or locations; financial or health data; biometric or genetic data; forms of government identification, such as social security numbers; criminal history)?** If so, please provide a description.

No

| **Collection Process** |
| --- |

**How was the data associated with each instance acquired?** Was the data directly observable (*e.g.* raw text, movie ratings), reported by subjects (*e.g.* survey responses), or indirectly inferred/derived from other data (*e.g.* part-of-speech tags, model-based guesses for age or language)? If data was reported by subjects or indirectly inferred/derived from other data, was the data validated/verified? If so, please describe how.

The raw video data, which is directly observable, was procured from the publicly accessible datasets (Camouflaged Animals Dataset (CAD) (Pia Bideau, 2016), Moving Camouflaged Animals (MoCA) (Lamdouar et al., 2020), MoCA-Mask (Cheng et al., 2022b), Marine Video Kit (MVK) (Truong et al., 2023) and crawled video from internet) as shown in Table 1 and Section 3 in the main paper. We utilized an annotation tool from (Zheng et al., 2023a), which is heavily based on Segment Anything Model (SAM) (Kirillov et al., 2023) for mask initialization and bounding box and XMem (Cheng & Schwing, 2022) for mask and bounding box propagation. We utilized the RAFT method (Teed & Deng, 2020) to produce an accurate estimated optical flow and refined it using Algorithm 1. To construct referring expression annotations, we utilized GPT-4V (OpenAI, 2023) to create a concise description for flying and terrestrial animals, and MarineGPT (Zheng et al., 2023b) for aquatic animals.

**What mechanisms or procedures were used to collect the data (*e.g.* hardware apparatus or sensor, manual human curation, software program, software API)?** How were these mechanisms or procedures validated?

The videos were downloaded in accordance with the official guidelines for data access of other datasets. For additional videos, we manually curated from the internet. See Section 3 in the main paper for a more detailed explanation.

**If the dataset is a sample from a larger set, what was the sampling strategy (*e.g.* deterministic, probabilistic with specific sampling probabilities)?**

We used all samples from the published datasets. So, there is no sampling strategy.

**Who was involved in the data collection process (*e.g.* students, crowd-workers, contractors) and how were they compensated (*e.g.* how much were crowd-workers paid)?**

The authors were involved in the data collection process. No crowd-workers were involved during the data collection process.

**Over what timeframe was the data collected? Does this timeframe match the creation timeframe of the data associated with the instances (*e.g.* recent crawl of old news articles)?** If not, please describe the timeframe in which the data associated with the instances was created.

The original videos within the published datasets were collected across various occasions spanning from 2011 to 2022. As for the CamoVid60K, the new videos were collected over several sprints during the first half of 2024.

**Were any ethical review processes conducted (*e.g.* by an institutional review board)?** If so, please provide a description of these review processes, including the outcomes, as well as a link or other access point to any supporting documentation.

No

**Does the dataset relate to people?** If not, you may skip the remaining questions in this section.

No

**Did you collect the data from the individuals in question directly or obtain it via third parties or other sources (*e.g.* websites)?**

NA

**Were the individuals in question notified about the data collection?** If so, please describe (or show with screenshots or other information) how notice was provided, and provide a link or other access point to, or otherwise reproduce, the exact language of the notification itself.

NA

**Did the individuals in question consent to the collection and use of their data?** If so, please describe (or show with screenshots or other information) how consent was requested and provided, and provide a link or other access point to, or otherwise reproduce, the exact language to which the individuals consented.

NA

**If consent was obtained, were the consenting individuals provided with a mechanism to revoke their consent in the future or for certain uses?** If so, please provide a description, as well as a link or other access point to the mechanism (if appropriate).

NA

**Has an analysis of the potential impact of the dataset and its use on data subjects (*e.g.* a data protection impact analysis) been conducted?** If so, please provide a description of this analysis, including the outcomes, as well as a link or other access point to any supporting documentation.

NA

---

| Preprocessing/cleaning/labeling |
|:---:|

**Was any preprocessing/cleaning/labeling of the data done (*e.g.* discretization or bucketing, tokenization, part-of-speech tagging, SIFT feature extraction, removal of instances, processing of missing values)?** If so, please provide a description. If not, you may skip the remainder of the questions in this section.

There was no preprocessing done on the videos, and we only did the frame extraction from the videos.

**Was the "raw" data saved in addition to the preprocessed/cleaned/labeled data (*e.g.* to support unanticipated future uses)?** If so, please provide a link or other access point to the "raw" data.

The raw data in our CamoVid60K dataset is video. However, all methods will extract videos into frames, so we only provide the extracted frames in our CamoVid60K dataset.

**Is the software used to preprocess/clean/label the instances available?** If so, please provide a link or other access point.

We used the FFmpeg library to extract the frames. The packages, executable files, and sources for Windows, macOS, Linux, or build from source are available in their official website.

---

| Distribution |
|:---:|

**Will the dataset be distributed to third parties outside of the entity (*e.g.* company, institution, organization) on behalf of which the dataset was created?** If so, please provide a description.

The dataset will be made publicly available and can be used for non-commercial research and educational purposes under the CC-BY-4.0 license.

**How will the dataset be distributed (*e.g.* tarball on website, API, GitHub)** Does the dataset have a digital object identifier (DOI)?

The dataset will be distributed at our CamodVid60K website (we will update our website later) upon acceptance to preserve anonymization.

**When will the dataset be distributed?**

The complete dataset will be made available upon the acceptance of the paper before the camera-ready deadline.

**Will the dataset be distributed under a copyright or other intellectual property (IP) license, and/or under applicable terms of use (ToU)?** If so, please describe this license and/or ToU, and provide a link or other access point to, or otherwise reproduce, any relevant licensing terms or ToU, as well as any fees associated with these restrictions.

CamoVid60K dataset will be publicly released under the CC-BY-4.0 license, which allows direct public use of the video/frames and annotation data for non-commercial research and educational purposes.

**Have any third parties imposed IP-based or other restrictions on the data associated with the instances?** If so, please describe these restrictions and provide a link or other access point to, or otherwise reproduce, any relevant licensing terms, as well as any fees associated with these restrictions.

No

**Do any export controls or other regulatory restrictions apply to the dataset or to individual instances?** If so, please describe these restrictions and provide a link or other access point to, or otherwise reproduce, any supporting documentation.

No

| Maintenance |
|:---:|

**Who will be supporting/hosting/maintaining the dataset?**

The authors of the paper will be maintaining the dataset, pointers to which will be hosted on our CamodVid60K website (we will update our website later) along with the guideline for download and preprocessing if needed.

**How can the owner/curator/manager of the dataset be contacted (*e.g.* email address)?**

We will post the contact information on our website, primarily contact through email.

**Is there an erratum?** If so, please provide a link or other access point.

In the future, we will host an erratum on our CamodVid60K website (we will update our website later) to host any approved errata suggested by the authors or the video research community.

**Will the dataset be updated (*e.g.* to correct labeling errors, add new instances, delete instances)?** If so, please describe how often, by whom, and how updates will be communicated to users (*e.g.* mailing list, GitHub)?

Yes, we plan to host an erratum publicly. There are no specific plans for a v2 version, but there seem to be plenty of opportunities for exciting future dataset work based on CamoVid60K.

**If the dataset relates to people, are there applicable limits on the retention of the data associated with the instances (*e.g.* were individuals in question told that their data**

**would be retained for a fixed period of time and then deleted)?** If so, please describe these limits and explain how they will be enforced.

No.

**Will older versions of the dataset continue to be supported/hosted/maintained?** If so, please describe how. If not, please describe how its obsolescence will be communicated to users.

N/A There are no older versions at the current moment. All updates regarding the current version will be communicated via our website.

**If others want to extend/augment/build on/contribute to the dataset, is there a mechanism for them to do so?** If so, please provide a description. Will these contributions be validated/verified? If so, please describe how. If not, why not? Is there a process for communicating/distributing these contributions to other users? If so, please provide a description.

Contributions will be made possible using comment functions in our CamodVid60K website (we will update our website later). The CamoVid60K team will verify any new contributions before publishing them on our website, and then we will host any approved errata suggested by the video research community.

