# OpenReview forum: "CamoVid60K: A Large-Scale Video Dataset for Moving Camouflaged Animals Understanding"
_ICLR.cc/2025/Conference — Submitted to ICLR 2025_

### Official Review · Reviewer_RR77 · 2024-10-29

**Soundness:** 2
**Presentation:** 3
**Contribution:** 2
**Rating:** 3
**Confidence:** 4

**Summary:**

The authors have introduced a comprehensive dataset of camouflaged animal videos, which is substantial in scale and includes various types of annotations. This dataset holds the potential to contribute to the tasks of camouflaged animal segmentation and understanding. They have also proposed a pipeline for video camouflaged animal segmentation that integrates existing methods, and conducted experiments on the proposed dataset. This approach demonstrates performance comparable to some existing methods. Additionally, the authors have experimented with other related tasks in camouflaged animal video understanding, yielding some results.

**Strengths:**

- Building on existing segmentation tasks, the authors introduce the concept of camouflaged scene understanding, and provide a thorough analysis of related work in this field. The proposed camouflaged animal video dataset includes motion annotations and referring expression annotations, contributing to the expansion of research in this domain. The dataset also offers dense annotations for each frame, making it the most annotated dataset in the current field.
- The article is well-written and presented, with key parts in bold and aesthetically pleasing statistical charts. The supplementary material also has a well-constructed display interface.

**Weaknesses:**

The main portion of the article focuses on video camouflaged object segmentation, where the novelty and contribution of the work are somewhat questionable.
1. There are concerns about the legitimacy of using existing dataset masks. The proposed dataset includes a significant portion derived from existing datasets, some of which already have mask annotations (e.g., moca-mask). While the video material is sourced from the public domain, the mask annotations belong to the authors of the previous work. Mask propagation from sparse masks to dense masks is not an overly challenging task, as the article mentions using some models for mask propagation. Could the authors clarify whether they used masks annotated by previous authors for propagation, and how exactly they did so? Additionally, I would like to know how many video clips in the dataset were collected by the authors themselves, as this pertains to the dataset's contribution.

2. There is a question about the necessity of dense annotations. I appreciate the authors' effort in data annotation. However, in Section 2.2, they mention that existing datasets' strategy of annotating one frame every five frames results in insufficient data. In practical use, whether for detection or segmentation, the necessity of dense annotations is not evident, and models may struggle to process data at 30 frames per second. The dataset proposed in this paper claims dense annotation as one of its main contributions, but the authors have not provided experiments or literature to demonstrate the usefulness of frame-by-frame annotation, which weakens the paper's innovative support. The authors should supplement relevant experiments, such as comparing the performance of models annotated every five frames versus frame-by-frame.

3. The use of perceptual camouflage score: The article mentions multiple times in Sections 3.1 and the appendix that the authors used the perceptual camouflage score proposed by Lamdouar et al. (2023) to quantify the effectiveness of animals' camouflage and filter candidate videos. While introducing this score for data filtering is innovative, this method requires annotated masks to be used, raising the question of how the authors filtered videos during the data collection phase. In Appendix A.2, the authors mention manually reviewing and initially obtaining 218 videos, with the final dataset also consisting of 218 video segments. Can I infer that no videos were filtered out during the second stage of using the perceptual camouflage score? Furthermore, in Figure 8, nine out of the ten image pairs shown were already present in Lamdouar’s original paper, which raises doubts about the authenticity of the perceptual camouflage score usage. Could the authors provide more evidence to support this?

4. The pipeline proposed in the article lacks strong innovation and contribution. The authors' pipeline uses an existing transformer architecture, but its performance in video camouflaged object detection and segmentation surpasses only unsupervised video methods or methods using only CNNs. This suggests the pipeline's limited novelty and contribution. If the pipeline is intended to be one of the main contributions of the paper, additional comparative experiments on MoCA-Mask are necessary.

While the paper's aspects on camouflaged animal video understanding are innovative, they lack sufficient discussion and experimental support.

- Hala Lamdouar, Weidi Xie, and Andrew Zisserman. The making and breaking of camouflage. InICCV, pp. 832–842, 2023.

**Questions:**

Could the authors respond to and revise the issues mentioned in the "Weaknesses" section, particularly the concerns regarding the use of the perceptual camouflage score.

---

> ### Author Response · Authors · 2024-11-14
> **To Reviewer RR77**
>
> **Q1** Legitimacy of Using Existing Dataset Masks and Dataset Contribution.
>
>  > **A1** As shown in Table 1, only CAD and MoCA-Mask datasets provided mask annotation, while MoCA dataset provided bounding box annotation. However, they only provided annotations for every 5 frames, resulting in only small amount of annotated frames (e.g. only 4,691 annotated frames of 22,939 frames from MoCA-Mask). In addition, the annotated mask from MoCA-Mask is image-based annotation, which is different from ours (json format as shown in Figure 3). Therefore, to ensure consistency and meet the dense (every frame) annotation requirements of our dataset, we re-annotated all videos sourced from existing datasets. Finnaly, in the Appendix A1, we mentioned that we collected 1,929 videos (including the videos from the existing datasets), and in Appendix A2, we manually reviewed and removed blurry or irrelevant videos (e.g. those with clearly visible animals), resulting in 218 videos with 62,774 frames.
>
>  **Q2** Necessity of Dense Annotations.
>
>  > **A2** Previous studies from general video object segmentation [13-15] and special domains (camouflaged or salient objects) [16-19] have shown that the dense or frame-by-frame annotations benefit model learning by capturing temporal dynamics more effectively. In addition, [18,19] proved that utilizing pseudo-label (because they don't have GT labels for every frame) can improve the performance of video approaches as they can leverage temporal information to suppress the label noises.
>  >
>  > [13] Zhang et al., Video Object Segmentation through Spatially Accurate and Temporally Dense Extraction of Primary Object Regions, CVPR 2013.
>  >
>  > [14] Perazzi et al., A Benchmark Dataset and Evaluation Methodology for Video Object Segmentation, CVPR 2016.
>  >
>  > [15] Xu et al., YouTube-VOS: A Large-Scale Video Object Segmentation Benchmark, ECCV 2018.
>  >
>  > [16] Wang et al., Video Salient Object Detection via Fully Convolutional Networks, T-IP 2018.
>  >
>  > [17] Chen et al., Progressively Complementarity-Aware Fusion Network for RGB-D Salient Object Detection, CVPR 2018.
>  >
>  > [18] Yan et al., Semi-Supervised Video Salient Object Detection Using Pseudo-Labels, ICCV 2019.
>  >
>  > [19] Cheng et al., Implicit Motion Handling for Video Camouflaged Object Detection, CVPR 2022.
>
>  **Q3** Concerns about the use of Perceptual Camouflage Score.
>
>  > **A3**
>  > * Sorry for confusion and we thank the reviewer to point out the Appendix A2. Our data filtering process include both manually review by human observation and perceptual camouflage score. In most of the cases, our observation align with the perceptual camouflage score. We will enhance this description in the revised version.
>  > * For the examples of Perceptual Camouflage Score in the supplementary, we used some same sets with Lamdouar’s original paper to show the illustration of low and high scores, and also verify the reported scores in Lamdouar’s original paper and our computed scores, which are mostly similar.
>
>  **Q4** Pipeline Innovation and Contribution.
>
>  > **A4** Our main contribution is a dataset which includes data, annotations, simple baseline for future research and benchmark. In addition, please prefer to the answer of Q2 of Reviewer nU6R for our discussion of the pipeline.

---

### Official Review · Reviewer_nU6R · 2024-10-31

**Soundness:** 2
**Presentation:** 3
**Contribution:** 2
**Rating:** 3
**Confidence:** 4

**Summary:**

This paper delves into the realm of understanding camouflaged animals through video analysis, a domain that has traditionally been explored at image and object levels. This study pioneers advancements by focusing on video and pixel-level comprehension. A new dataset sourced from the web is introduced, with annotations primarily derived from off-the-shelf models. The proposed baseline model integrates visual and motion encoders along with a mask decoder, all leveraging existing model architectures. Experimental results demonstrate the efficacy of this streamlined approach compared to state-of-the-art models on the new benchmark presented.

**Strengths:**

* The investigation into camouflaged animal understanding signifies a challenging and pivotal research avenue deserving increased attention within the academic community. This work represents a progressive step towards broadening the scope of this critical topic to encompass a wider array of species with richer data.
* The authors' dedication to curating a substantially larger dataset than what is currently available is commendable.
* The automated annotation process utilizing off-the-shelf models to identify objects at the image-level and propagate masks throughout the video stream proves to be an efficient pipeline, reducing both time and manual labor.

**Weaknesses:**

* The reliance on fully annotating the dataset with off-the-shelf deep learning models raises concerns about the thoroughness of the labeling process. While the authors mention refining outputs, the lack of details on this aspect questions the depth of understanding achieved in camouflaged animal analysis. This raises the issue of whether the combination of existing techniques adequately addresses the challenge at hand, potentially framing it more as an engineering problem rather than a research problem.
* The proposed baseline model provides limited insights to the field. Employing a combination of a visual encoder, motion encoder, and mask decoder, each sourced from established model architectures, which results in strong performance in camouflaged animal understanding, does not strike me as particularly innovative. This approach appears to be widely acknowledged in the realm of pixel-level video comprehension.
* An essential aspect that requires clarification is how these components are initialized. Whether they are trained from scratch or initialized with pretrained weights can significantly impact the model's performance, especially when working with limited data volumes. Training intricate models like Mask2Former from scratch with a small dataset can pose considerable challenges due to the complexity and the volume of data necessary for effective learning. Besdeis, if pretrained weights are employed in the model, it is crucial for the authors to disclose this information in their study to ensure fairness in comparative evaluations.
* The paper lacks a thorough exploration of the distinctive challenges inherent in camouflaged animal understanding. Addressing specific challenges tackled by the dataset and the proposed pipeline is essential. While the data sourcing strategy is unique, further differentiation is required for publication.
* The benchmarking methodology appears somewhat lacking, as it only considers methods published before 2022. To establish a more robust benchmark, the inclusion of the latest video object segmentation techniques with superior performance is necessary for a comprehensive evaluation.

**Questions:**

* The paper mainly focuses on identifying salient objects in scenes. It would be interesting to explore how the model handles situations with multiple instances of camouflaged animals.

---

> ### Author Response · Authors · 2024-11-14
> **To Reviewer nU6R**
>
> **Q1** Concerns about thoroughness in dataset labeling and research focus.
>
>  > **A1** We appreciate the reviewer’s concerns regarding the labeling process, and also the complement on an efficient pipeline, reducing both time and manual labor. We would like to clarify that while we rely on off-the-shelf models with human assistance (providing points as prompt for SAM model) to generate initial annotations, a rigorous manual refinement process is undertaken to ensure accuracy and relevance to camouflaged animal analysis. The refinement stage involves domain experts (the authors have been working on scene understanding field for more than a decade) validating and correcting the propagated masks (generated by XMem method) to mitigate errors and improve data quality.
>
>  **Q2** Perceived lack of innovation in the baseline model.
>
>  > **A2** While the architecture components are sourced from existing models, our contribution lies in their specific integration and optimization tailored to camouflaged animal understanding (subtle motion cues and low contrast between the animal and its surroundings). It is based on Mask2Former architecture and act as a baseline for both object detection and motion segmentation tasks. The main difference is that we extend Mask2Former, which takes only one RGB image as input to take both sequence of RGB image and optical flow. Then, instead of directly outputting the prediction from Decoders, we output both the mask and bounding box prediction from the aggregation of the motion features (from Pixel Decoder) and the mask embedding (from Transformer Decoder).
>
>  **Q3** Clarification on model initialization and training strategy.
>
>  > **A3** As mentioned in line 369, we adopt SINet-v2 architecture for our visual encoder. In detail, SINet-v2 utilized ResNet50 as backbone with weights pretrained on ImageNet dataset. We will explicitly include the ResNet50 backbone in the revised manuscript.
>
>  **Q4** Exploration of distinctive challenges in camouflaged animal understanding.
>
>  > **A4** As mentioned in lines 50-53, our dataset potentially can help deep learning methods for monitoring and surveillance of wildlife animals, such as the camera trap method. Then, with the captured data, we can further study biodiversity. From the reference works, Troscianko et al. discusses how isolating camouflaged animals from their backgrounds allows researchers to gather precise data on their behavior, habitat preferences, and how their camouflage strategies evolve over time. Yunqiu et al. [2] highlights how modeling the conspicuousness of camouflaged objects enhances our understanding of camouflage and animal evolution.
>
>  **Q5** Benchmarking methodology and inclusion of recent methods.
>
>  > **A5** Please prefer to our answer of Q2 of Reviewer JSBg above.
>
>  **Q6** The paper mainly focuses on identifying salient objects in scenes. It would be interesting to explore how the model handles situations with multiple instances of camouflaged animals.
>
>  > **A6** We think that there is a misunderstanding. Our paper focuses on camouflaged objects (animals), which is essentially the opposite of salient objects. Regarding the multiple instances of camouflaged animals, our model's current architecture based on Mask2Former, inherently supports scalability to multi-instance detection. We hypothesize that minor modifications, such as adding a dedicated module for instance separation or leveraging ensemble approaches, would enhance its multi-instance segmentation performance [11,12].
>  >
>  > [11] Cheng et al., "Mask2Former for Video Instance Segmentation," arXiv preprint 2112.10764, 2021.
>  >
>  > [12] Lei et al., "Mask-Free Video Instance Segmentation," CVPR, 2023.

---

> > ### Comment · Reviewer_nU6R · 2024-11-27
> >
> > Thank you for the comprehensive response from the authors, which addressed a significant portion of my concerns. Upon thorough consideration of the rebuttal and feedback from other reviewers, I am inclined to maintain my rating. I acknowledge that this paper shows promise in its trajectory but could benefit from further refinement in its specifics.

---

> > > ### Author Response · Authors · 2024-11-28
> > >
> > > Thank you for your response and if you have additional questions, we are happy to respond. In addition, as you mentioned above, we have addressed a significant portion of your concerns, so could you consider to raise your rating?

---

### Official Review · Reviewer_mugH · 2024-11-02

**Soundness:** 2
**Presentation:** 2
**Contribution:** 2
**Rating:** 5
**Confidence:** 5

**Summary:**

The paper introduces CamoVid60K, a comprehensive video dataset aimed at enhancing the study of camouflaged animals through improved segmentation and detection in videos. This dataset comprises 218 videos and 62,774 finely annotated frames, covering 70 distinct animal species. The authors detail the construction, annotation process, and the evaluation benchmarks established using state-of-the-art models. CamoVid60K stands out for its exhaustive annotation types—bounding boxes, masks, pseudo-optical flow, and referring expressions—and provides comparisons with existing datasets to illustrate its scale and uniqueness.

**Strengths:**

1. The dataset fills a clear research gap in video-based camouflaged animal detection and segmentation.

2. Thorough comparison with existing datasets and the benchmarking of state-of-the-art methods highlight the dataset’s relevance and potential for advancing the field.

**Weaknesses:**

My main concerns is that the paper focuses more on dataset development and benchmarking rather than presenting innovative new algorithms or techniques. Consequently, if the contribution to the field of VCOD is limited to proposing a dataset containing only 60k video frames, it may not be sufficient to consider the overall impact of the paper substantial. Moreover, I have noticed that the dataset proposed by the authors does not seem to facilitate advancements in other fields.

**Questions:**

Please see weaknesses.

---

> ### Author Response · Authors · 2024-11-14
> **To Reviewer mugH**
>
> **Question** The paper focuses more on dataset development and benchmarking rather than presenting innovative new algorithms or techniques. Consequently, if the contribution to the field of VCOD is limited to proposing a dataset containing only 60k video frames, it may not be sufficient to consider the overall impact of the paper substantial. Moreover, I have noticed that the dataset proposed by the authors does not seem to facilitate advancements in other fields.
>
>  > **Answer**
>  > * Our paper aim to fill the gap of lacking a diverse, large-scale, densely and accurately annotated video dataset of camouflaged animals. That's why the primary area of our submission is the datasets and benchmarks.
>  > * In addition, as shown in Table 1 and lines 87-100, our dataset **surpasses** all previous datasets in terms of the number of videos, frames, and species included. Unlike previous datasets that annotated *every 5 frames*, our dataset offers annotations for **every single frame**. Additionally, we provide **a wider variety of annotation types** (animal categories, bounding boxes, annotated masks, pseudo-label optical flow, referring expressions), making it a more effective benchmark for camouflaged animal video understanding tasks. Finally, our dataset supports **a broad range of downstream tasks**, including classification, detection, segmentation (semantic, referring, motion), and optical flow estimation.
>  > * Lastly, in our Conclusion section, we have discussed the impact on animal studies (lines 513-518). In detail, Troscianko et al. [7] explore the impact of camouflage on predator-prey interactions. Norouzzadeh et al. [8] demonstrates the effectiveness of deep learning in wildlife identification and monitoring. Beery et al. [9] and Fanny et al. [10] discuss the use of bounding boxes and deep learning in wildlife imagery for ecological research and conservation efforts.
>  >
>  > [7] Troscianko et al., "Quantifying camouflage: how to predict detectability from appearance." BMC Ecology and Evolution, 2017.
>  >
>  > [8] Norouzzadeh et al., "Automatically identifying, counting, and describing wild animals in camera-trap images with deep learning." PNAS, 2018.
>  >
>  > [9] Beery et al., "Recognition in terra incognita." ECCV, 2018.
>  >
>  > [10] Fanny et al., "DeepWILD: Wildlife Identification, Localisation and estimation on camera trap videos using Deep learning." Ecological Informatics, 2023.

---

### Official Review · Reviewer_JSBg · 2024-11-04

**Soundness:** 2
**Presentation:** 3
**Contribution:** 2
**Rating:** 5
**Confidence:** 4

**Summary:**

- The paper presents CamoVid60K, a comprehensive video dataset dedicated to studying camouflaged animals.

- The dataset contains 218 videos, with 62,774 annotated frames, covering 70 animal categories.

- The dataset contains annotations of various forms, for example, bounding box, segmentation masks, classification, and optical flow.

- The authors have also proposed a simple pipeline for camouflaged animal detection and segmentation with the dataset.

**Strengths:**

- The investigated problem of camouflage animal detection is interesting.

- The contribution of dataset is good, though the construction procedure involves quite some manual effort, thus limiting its scalability.

- The writing is clear, especially on the dataset curation part.

**Weaknesses:**

- The dataset is of relative small scale.

- The compared models are quite old, for example, MG was published in 2021, SLT-Net was published in 2022. In addition, as far as I know, MG was trained with self-supervised learning, thus the comparison is not that fair.

- The contribution on the architecture design is minor, and trained on the video dataset with complete supervised learning, which limits the value of proposed method, as it is not scalable, as has been done in previous work, either with synthetic data, or self-supervised learning.

**Questions:**

Could the authors evaluate more new models ? for example, [1], which has already been cited in the paper.

[1] Xie et al. Segmenting moving objects via an object-centric layered representation, NeurIPS 2022.

And I'm sure there are more new models.

---

> ### Author Response · Authors · 2024-11-14
> **To Reviewer JSBg**
>
> **Q1** The dataset is of relative small scale.
>
>  > **A1** As shown in Table 1 and lines 87-100, our dataset **surpasses** all previous datasets in terms of the number of videos, frames, and species included. Unlike previous datasets that annotated *every 5 frames*, our dataset offers annotations for **every single frame**. Additionally, we provide **a wider variety of annotation types** (animal categories, bounding boxes, annotated masks, pseudo-label optical flow, referring expressions), making it a more effective benchmark for camouflaged animal video understanding tasks. Our dataset supports **a broad range of downstream tasks**, including classification, detection, segmentation (semantic, referring, motion), and optical flow estimation.
>
>  **Q2** The compared models are quite old, for example, MG was published in 2021, SLT-Net was published in 2022. In addition, as far as I know, MG was trained with self-supervised learning, thus the comparison is not that fair.
>
>  > **A2**
>  > * Regarding the age of the models: The selection of MG (2021) and SLT-Net (2022) was based on their relevance as established benchmarks in the field. These models represent widely acknowledged milestones for the specific tasks we address, and their inclusion provides context for evaluating the comparable performance introduced by our method. In addition, there are many recent published methods [1-5] and under review method [6], but to provide the most direct comparison, we focused on models with publicly available implementations (no available source code for [1,4-6]) and standardized evaluations (it is unfair to compare with SAM-based methods [2,3]).
>  > * On fairness of the comparison with MG (self-supervised learning): we acknowledge the differences in training paradigms between MG (self-supervised learning) and our method (fully supervised). The inclusion of MG in our comparisons was not to claim a direct superiority but rather to provide a broader perspective on how our method performs relative to diverse approaches, including self-supervised ones. To ensure fairness, we will explicitly highlight these differences in the revised manuscript, emphasizing that MG’s performance reflects its specific training paradigm.
>  >
>  > [1] Yu et al., Tokenmotion: Motion-Guided Vision Transformer for Video Camouflaged Object Detection VIA Learnable Token Selection, ICASSP 2024.
>  >
>  > [2] Meeran et al., SAM-PM: Enhancing Video Camouflaged Object Detection using Spatio-Temporal Attention, CVPRW 2024.
>  >
>  > [3] Hui et al., Endow SAM with Keen Eyes: Temporal-spatial Prompt Learning for Video Camouflaged Object Detection, CVPR 2024.
>  >
>  > [4] Hui et al., Implicit-Explicit Motion Learning for Video Camouflaged Object Detection, T-MM 2024.
>  >
>  > [5] Bideau et al., The Right Spin: Learning Object Motion from Rotation-Compensated Flow Fields, IJCV 2024.
>  >
>  > [6] Zhang et al., Explicit Motion Handling and Interactive Prompting for Video Camouflaged Object Detection, arXiv 2024.
>
>
>
>  **Q3** The contribution on the architecture design is minor, and trained on the video dataset with complete supervised learning, which limits the value of proposed method, as it is not scalable, as has been done in previous work, either with synthetic data, or self-supervised learning.
>
>  > **A3** Our proposed method is a simple pipeline which aims to provide a baseline to further explore our dataset on both motion detection and segmentation tasks of camouflaged animals.
>
>
>  **Q4** Could the authors evaluate more new models ?
>
>  > **A4** Please prefer to our answer of Q2 above.

---

### Author Response · Authors · 2024-11-14
**Rebuttal by Authors**

## Overall response
Dear reviewers,

We thank all the reviewers for their constructive feedback.

**All reviewers** have recognized the novelty and value of our dataset, highlighting its significant potential for applications in computer vision, particularly within wildlife studies. All reviewers have recognized the novelty and value of our dataset, highlighting its significant potential for applications in computer vision, particularly within wildlife studies.

**Reviewer JSBg** acknowledged our dataset as a comprehensive resource for camouflaged animal detection and segmentation, noting the interesting problem it addresses. They commended the clarity of our writing, especially regarding the dataset curation part.

**Reviewer mugH** commended the quality of our dataset, emphasizing that it fills a clear research gap in video-based camouflaged animal detection and segmentation. They appreciated the thorough comparisons with existing datasets and the benchmarking of state-of-the-art methods, highlighting the dataset's relevance and potential for advancing the field.

**Reviewer nU6R** appreciated the broad impact of our dataset, especially its substantial size compared to existing resources, and recognized our dedication in curating it. They highlighted the efficiency of our automated annotation process using off-the-shelf models, which reduces time and manual labor. They viewed our work as a progressive step toward broadening the scope of camouflaged animal understanding to encompass a wider array of species with richer data.

**Reviewer RR77** acknowledged our clear contribution in introducing the concept of camouflaged scene understanding and providing a thorough analysis of related work. They noted that our dataset includes various annotation types—such as motion annotations and referring expression annotations—marking it as the most annotated dataset in the current field. They also praised the well-written and presented article, including the aesthetically pleasing statistical charts and the well-constructed supplementary material.

We will response to the reviewers separately below. Could the reviewers please let us know if there are other concerns after reading our responses? We will be happy to address any further concerns.

Regards,

The authors

---

### Meta-Review · Area_Chair_FF2D · 2024-12-21

**Metareview:**

This paper presents a new (unified over some previously existed) dataset for Camouflaged Animals Understanding in videos. The dataset consists of 218 videos with 62,774 frames and mask annotations are provided densely at 5fps. A simple baseline is also provided and compared with previous approaches.

Although all reviewers appreciate the novelty and the importance of the problem, all reviewers still have concerns which result in low ratings. The main concerns include: (i) the small size of the dataset; (ii) the usage of the off-the-shelf models during annotation; (iii) the lack of novelty in the proposed baseline; (iv) missing comparison with latest methods (after 2022).

The authors provided response to reviewers but all reviewers agree that the paper need to address those concerns before it is ready for publication. AC reads all reviews and responses and agrees with the reviewers. Thus, AC recommends a rejection. AC recommends the authors to address the concerns provided by the reviewers and re-submit the work to future conferences.

**Additional Comments On Reviewer Discussion:**

As mentioned above, the main concerns from the reviewers are: (i) the small size of the dataset; (ii) the usage of the off-the-shelf models during annotation; (iii) the lack of novelty in the proposed baseline; (iv) missing comparison with latest methods (after 2022).

Although the paper has the potential, the area chair agrees with the reviewers that the paper is not yet ready for publication at its current form. Thus recommends a rejection. However, AC encourages the authors to address the concerns provided by the reviewers and re-submit the work to future conferences.

---

### Decision · Program_Chairs · 2025-01-22

Reject